**Data Availability Statement:** http://ctxses.saude.sp.gov.br/ https://www.irodat.org/.

# Model for establishing a new liver transplantation center through mentorship from a university with transplantation expertise

Rafael Soares Pinheiro[1], Wellington Andraus[1], Fernando Gomes Romeiro[2]*, Rodrigo Bronze de Martino[1], Liliana Ducatti[1], Rubens Macedo Arantes[1], Leonardo Pelafsky[2], Claudia Nishida Hasimoto[2], Fabio da Silva Yamashiro[2], Lucas Souto Nacif[1], Luciana Bertocco de Paiva Haddad[1], Vinicius Rocha Santos[1], Daniel Reis Waisberg[1], Matheus Fachini Vane[3], Joel Avancini Rocha-Filho[3], Walmar Kerche de Oliveira[2], Luiz Augusto Carneiro-D'Albuquerque[1]

1 Liver and Digestive Organs Transplantation Division, Gastroenterology Department, Clinical Hospital of São Paulo University - HCFMUSP, São Paulo Faculty of Medicine, Universidade de São Paulo - USP, São Paulo, São Paulo, Brazil, 2 Gastroenterology Division, Botucatu Faculty of Medicine, Universidade Estadual Paulista Júlio de Mesquita Filho – UNESP, São Paulo, São Paulo, Brazil, 3 Anesthesiology Division and Anesthesiology Laboratory LIM08, Surgery Department, Clinical Hospital of São Paulo University - HCFMUSP, São Paulo Faculty of Medicine, Universidade de São Paulo - USP, São Paulo, São Paulo, Brazil

* fgromeiro@gmail.com, fernando.romeiro@unesp.br

## Abstract

### Background

Setting up new liver transplant (LT) centers is essential for countries with organ shortages. However, good outcomes require experience, because LT learning depends on a high number of surgeries. This study aims to describe how a new center was set up from a partnership between the new center and an experienced one. The step-by-step preparation process, the time needed and the results of the new center are depicted.

### Material and methods

The mentoring process lasted 40 months, in which half of the 52 patients included on the transplant list received LT. After the mentorship, a 22-month period was also analyzed, in which 46 new patients were added to the waiting list and nine were operated on.

### Results

The 30-day survival rates during (92.3%) and after (66.7%) the partnership were similar to the other LT centers in the same region, as well as the rates of longer periods. The waiting time on the LT list, the characteristics of the donors and the ischemia times did not differ during or after the mentorship.

**Funding:** This study was funded by São Paulo Research Foundation (FAPESP; grant no. 2017/25592-9). FGR received funding from Conselho Nacional de Desenvolvimento Científico e Tecnológico (CNPq).

**Competing interests:** The authors declare that they have no conflicts of interest to declare.

## Conclusion

The partnership between universities is a suitable way to set up LT centers, achieving good results for the institutions and the patients involved.

## Introduction

Liver transplantation (LT) is a key surgical treatment for several liver disorders. The discovery of calcineurin inhibitors has enabled the expansion of transplantation procedures throughout the world. Brazil was a pioneer in solid organ transplants in Latin America, having expanded the country's transplantation program since the 1990s with the implementation of the Unified Health System, named 'Sistema Único de Saúde' or 'SUS' (in Portuguese) [1, 2]. Through its public health system, Brazil currently performs the largest number of transplants in the world [3]. It is the sixth largest country in the world, with 208,494,900 inhabitants. In 2019, 2,245 LTs were performed in Brazilian centers. However, it represents only 43% of the number needed to empty the country's waiting list [4].

São Paulo, the most populous Brazilian state, is located in the southeast region of the country, and has a territorial extension of 248,209.43 km$^2$ and a population of 45,538,936 inhabitants (21.8% of the country) [4]. Although the state's target number of LT in 2019 was 1,138, only 699 procedures (61.4%) were performed [5]. Botucatu, a city located in the countryside of São Paulo state, is located 250 km away from other major centers, namely the state capital and other large cities such as Ribeirão Preto and Campinas.

The number of tertiary hospital centers that are capable of performing solid organ transplantations has increased, accompanied by the efficiency of the procedure, thus improving the organ donation rates [6]. To achieve these benefits, the distance between the recipient, the donor and the transplantation center cannot be excessively long, in order to enable operating on potential donors, transporting the organs and implanting them in the recipients within a proper time frame. However, starting a new transplant program involves high costs, mainly for hiring skilled professionals. Another problem is the initial learning curve, while the successful rates are not as high as expected [7].

A way to reduce these problems when starting a new transplant center is a period of mentoring, during which an institution with expertise in the field participates in and supervises the transplantations. A partnership between universities is even more advantageous because it allows the exchanging of students, knowledge and experiences.

The Clinical Hospital at the Botucatu Faculty of Medicine (HCFMB) was the new center set up during the mentorship. It is a reference hospital located in the city of Botucatu, with almost 500 beds and performing about 800 monthly surgeries [8]. The HCFMB has an Organ Procurement Organization (OPO) unit comprising 52 cities and 1.6 million inhabitants [4]. Since its first kidney transplant was performed in 1987, the hospital has been the site of more than 1,500 transplants. In 2003, the institution held the first LT; however, until 2013 only 12 LTs had been done, a result that did not achieve the expected outcomes. Consequently, the problems found during the learning curve led the team to discontinue the LT program in that year.

To restart the program in 2015, a partnership was signed between HCFMB and the Digestive Organs Transplantation Division of the Clinical Hospital of the São Paulo University (HCFMUSP). HCFMUSP was the national pioneer in this procedure in 1968 and performed the first living-donor LT in the world in 1985 [1]. In addition, it is the institution with the

highest number of LTs in Brazil. HCFMUSP has done more than 4,000 LTs and is the Brazilian hospital with the highest number of LTs annually in recent years [5].

This study aims to demonstrate the results of a partnership between two university hospitals to set up a new LT program, showing the step-by-step process, the time needed and the results obtained in the new transplantation center.

## Material and methods

A mentorship between November 2015 and March 2019 was given by HCFMUSP in order to start an LT program at HCFMB. The experienced center provided assistance in organizing the waiting list, the outpatient follow-up before and after transplantation, the organ procurement and the surgeries. The new center already had two experienced hepatologists and three digestive-tract surgeons skilled in complex surgical procedures and trained in abdominal organ transplantation, but with insufficient LT experience at that time. The new center also had professionals trained in crucial medical specialties for LT, such as pathology, radiology, infectious diseases and intensive-care medicine, thus composing a multidisciplinary team. Before starting the program, the experienced center made local visits to check the facilities and the infrastructure available, including the surgical instruments, equipment and supplies needed to perform transplantations. The new center bought all the requested instruments, such as vascular clamps and scissors.

Members of the two institutions evaluated the potential receivers through online meetings. Once a month, a surgeon from the experienced center assessed the potential recipients during outpatient appointments with the new team, preparing patients to be included on the waiting list. The preoperative preparation followed the assessment protocols of the experienced center. The decision to accept potential donors was made by the HCFMUSP medical team. During the first phase of the mentoring program, the experienced center's surgical team performed the organ procurement. The procurement team, accompanied by an HCFMB surgeon, performed the back-table surgeries for graft preparation.

An HCFMUSP team composed of a scrub nurse, surgeons and anesthesiologists participated in the recipient's surgery as tutors, always accompanied by an HCFMB surgeon. In the first months, the HCFMB surgeon performed parts of the transplantation, such as the hepatic vein anastomosis, the hepathectomy and other important tasks, which were frequently alternated between HCFMB and HCFMUSP surgeons, encouraging them to share the main tips and concerns. During this period, the experienced surgeons guided the new center team to do the procedures in the best way they knew. As soon as the mentors observed that HCFMB surgeons had done all of these parts safely and rapidly, new ones were shared, until the entire surgery could be carried out by the new team, always under supervision by an HCFMUSP surgeon.

The adopted surgical technique was similar to the ones performed at the experienced center. The following strategies were applied to avoid significant hemodynamic alterations in the recipient:

1. Ligation of the hepatic hilum structures followed by temporary portocaval shunt;

2. Inferior vena cava preservation (piggyback technique);

3. Suture of the ostium of the right hepatic vein;

4. Clamping only the anterior wall of the inferior vena cava to open the ostia of the middle and left hepatic veins, with lateral extension in the vena cava wall;

5. Repositioning of the vascular clamp at the end of the vena cava anastomosis to completely release the vena cava flow before the end of the temporary portocaval shunt.

The remaining LT steps were maintained without significant modifications.

One of the surgeons from the experienced center continued in the HCFMB during the initial postoperative period, remaining available for an eventual surgical re-approach in the first two days after the LT. Moreover, medical care and immunosuppression were guided through discussions between the two teams by phone. In a second phase, after August 2018, the new center team performed the organ procurement and the back-table surgeries.

On March 2019, the partnership program was completed and HCFMB started to perform LT without assistance. The data from this second period were compiled until January 2021. The data from patients transplanted by other teams were obtained through the Transplantation Center system of the São Paulo State Health Secretary [9]. Transplanted patients with allocation priority, such as acute liver failure, were not included in this analysis because the surgeries in these cases were not performed in the new center.

## Statistical analysis

Categorical variables were reported as number of cases and percentages, and were analyzed using the Chi-square test. Continuous variables that did not present normal distribution were expressed as medians followed by minimum and maximum values, and were analyzed by the Mann-Whitney test. The survival curves of the included patients were expressed using the Kaplan-Meier method. The significance level established for the analysis was 0.05. Statistical analyses were performed using the software SPSS 25.0 (SPSS, Chicago, IL).

## Ethics

This study was approved by the Ethics Committee of the São Paulo University and was conducted in accordance with the Declaration of Helsinki of 1996. Since the project was retrospective and observational, the need for consent was waived by the ethics committee.

## Results

The mentoring program lasted 40 months. In this time span, 52 patients were included on the transplant list. Their demographic characteristics are shown in Table 1. All the assessed variables were similar between the groups. Twenty-six transplants were performed, 14 patients died while on the waiting list, 7 patients were removed from the list, 4 patients were still on the waiting list and 1 patient was transferred to another center. Postoperative survival rates were 92.3% in 30 days, 88.5% in 1 year and 81.1% in 3 years. In the same period, transplant centers in the countryside of São Paulo state (Regional 2) performed 692 transplants. The actuarial survival rates of HCFMB and Regional 2 were 81.09% and 56.1%, respectively (Figs 1 and 2), with no statistical difference between them (p> 0.5).

After the mentorship, a 22-month period was analyzed, in which 46 new patients were added to the waiting list. The patients' characteristics are shown in Table 1. During this time span, 9 LTs were performed, 16 patients died on the waiting list, 13 patients are still waiting and 10 were removed from the list. The 30-day survival rate was 66.7%.

Data on donors and ischemia time are shown in Table 2. The comparison between the two groups did not show significant differences. None of the patients receiving a transplant at the new center had non-functioning primary graft, arterial thrombosis or needed re-transplantation.

## Discussion

The current study presents a successful strategy to set up an LT program through a mentoring period, during which an institution with expertise in this area helped a new center to achieve

**Table 1. Characteristics of the transplant patients at the new center during and after the partnership.**

| Variables | During the partnership | After the partnership | P |
|---|---|---|---|
| Listed patients | 52 | 48 | - |
| Age (range) | 58 (34–70) | 55 (20–72) | 0.907 |
| Male (%) | 42 (80.7) | 36 (75) | 0.484 |
| Liver diseases (%) | | | |
| Viral hepatitis | 25 (48) | 15 (13.1) | 0.086 |
| Alcohol abuse | 14 (26.9) | 10 (20.8) | 0.476 |
| NASH | - | 12 (25) | - |
| Other | 13 (25) | 11 (22.9) | 0.807 |
| BMI (range) | 27 (13–39) | 28 (17–42) | 0.578 |
| Child-Pugh (%) | | | |
| A | 23 (44.2) | 15 (31.3) | 0.181 |
| B | 10 (19.2) | 12 (25) | 0.486 |
| C | 19 (36.6) | 21 (43.7) | 0.462 |
| HCC (%) | 27 (51.9) | 19 (39.6) | 0.216 |
| O blood type (%) | 27 (51.9) | 25 (52.1) | 0.987 |
| A blood type (%) | 18 (34.6) | 18 (37.5) | 0.763 |
| AB blood type (%) | 3 (5.8) | 1 (2.1) | 0.347 |
| B blood type (%) | 4 (7.7) | 4 (8.3) | 0.856 |
| MELD (range) | 12 (7–36) | 17 (7–35) | 0.174 |
| Days on the liver transplantation list (range) | 219 (5–352) | 248 (24–328) | 0.226 |
| Days awaiting for liver transplantation before death (range) | 63.5 (2–380) | 80 (3–713) | 0.417 |

Values presented as median (minimum-maximum) or medians and percentage. BMI: body mass index, in kg/m$^2$. Child-Pugh: Child-Pugh classification. MELD: Model for End-Stage Liver Disease; NASH: non-alcoholic steatohepatitis.

good results. Despite the fact that the new center was a reference hospital that started performing transplantations in 1987, the institution had not been performing LT since 2013. Aiming to continue providing care to patients in this region, the partnership was signed establishing a mentorship provided by the experienced center. The number of LTs done during the 40-month partnership was far higher than the one performed between 2003 and 2013, and achieved better results compared to the centers in the same region.

Brazil has the largest transplant program through its public health system in the world. Ninety-five percent of more than 26 thousand solid organ transplantations per year are funded by SUS, the Brazilian public health service [3, 10]. According to the International Registry of Organ Donation and Transplants, Brazil ranks second among all countries in the highest absolute number of LTs. On the other hand, these numbers result in only 10.8 transplants per million population (pmp) [11], making the country 23rd in the international ranking of LT pmp. Hence, there is no doubt that Brazil needs to expand LT, but the access to transplant centers is a barrier, especially for the deprived population [12]. New LT centers must be created to serve broad areas, especially in the northern and center-west regions [13]. Nevertheless, it also requires the expansion of LT programs in intensely populated regions, such as those in the southern and southeastern regions.

The southeastern region is the most populous in Brazil, with 87 inhabitants per Km$^2$ [4] and the highest concentration of transplant centers [5]. Conversely, LT is not fully accessible because the transplant centers are concentrated in a few major cities. As a result, people who live far away from these cities have to be referred to them to have access to LT [14]. This

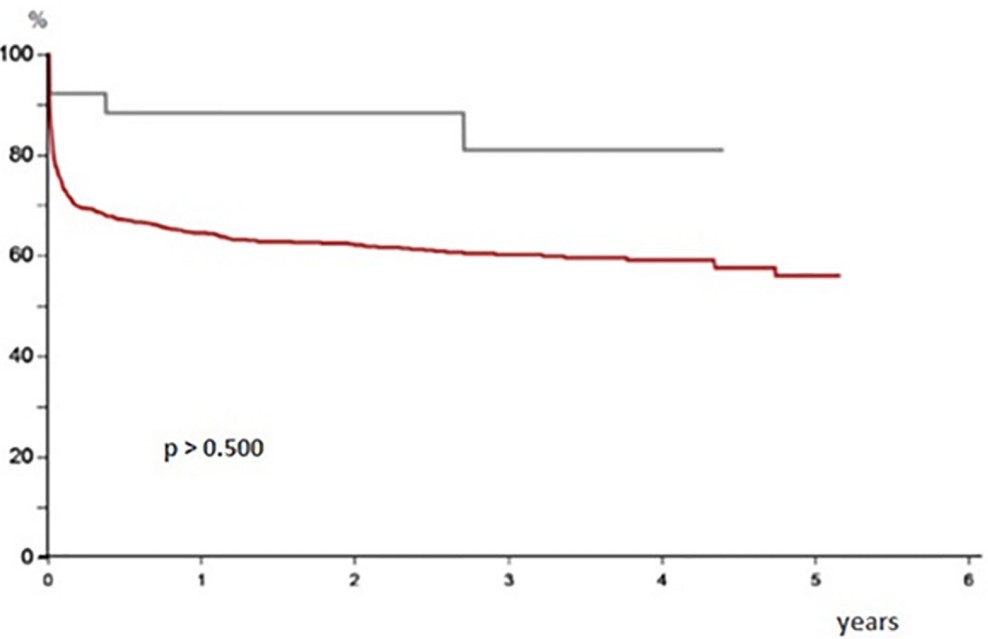

**Fig 1. Survival of patients operated on at the new center during the partnership (plotted in gray), compared to the survival of patients operated on at other transplant centers outside of the São Paulo State capital (plotted in red), in the same time span.**

problem is even worse for patients from the poorer cities that need specialized care [15]. Furthermore, LT requires adequate logistics so that the organ procurement team can access the deceased donors in time, especially in large countries with restricted air and rail transportation [2]. These data reinforce the need to expand the number of transplant centers, even in the southern and southeastern regions of the country.

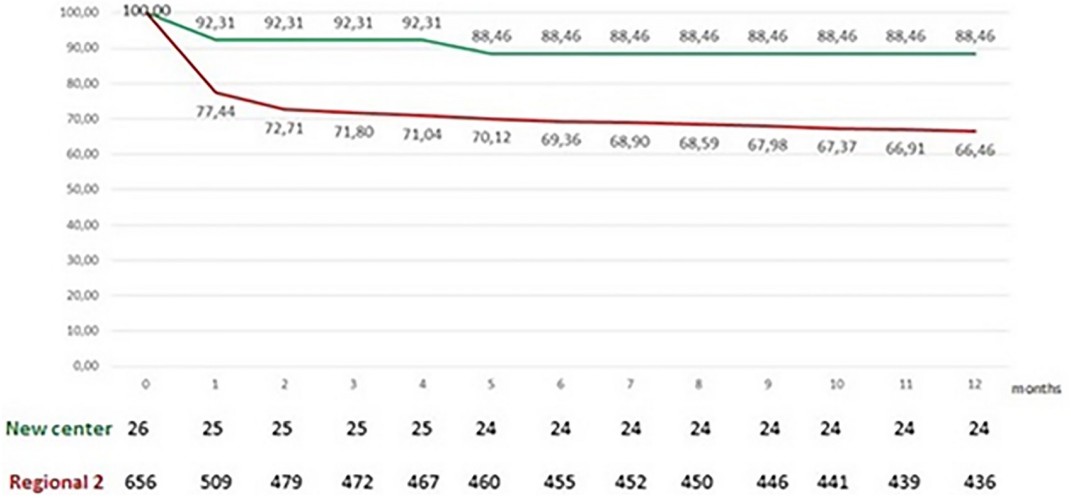

**Fig 2. Survival of patients operated on at the new center during the first year of the partnership (plotted in green), compared to that of patients operated on at other transplant centers outside of the São Paulo state capital (Regional 2, plotted in red).**

**Table 2. Liver transplants carried out in the new center during and after the partnership.**

| Variables | During the partnership | After the partnership | p |
|---|---|---|---|
| Transplants | 26 | 9 | - |
| Initial MELD | 29 (20–30) | 29 (24–31) | 0.896 |
| Donor age (range) | 39 (15–66) | 23 (16–53) | 0.126 |
| Donor BMI (range) | 24 (14–33) | 23 (21–25) | 0.173 |
| Donor days in ICU (range) | 4 (1–17) | 3 (1–22) | 0.726 |
| Brain death causes (%) | | | |
| Vascular | 14 (53.8) | 7 (77.7) | 0.206 |
| Trauma | 9 (34.6) | 2 (44.3) | 0.49 |
| Others | 3 (11.5) | - | - |
| Cold ischemia time (h:min, range) | 8:13 (5:49–13:50) | 8:24 (6:30–12:19) | 0.555 |
| Warm ischemia time (min, range) | 30 (20–47) | 30 (30–40) | 0.748 |

Values presented as median (minimum-maximum) or medians and percentage. BMI: body mass index, in kg/m$^2$. ICU: intensive care unit. MELD: Model for End-Stage Liver Disease.

The patients listed for LT in the present study had similar characteristics before and after the end of the partnership. Viral hepatitis was the most common etiology of liver disease and there was a clear male predominance, with a median age of 55–58 years. It is noteworthy that the median waiting time was over 200 days, thus demonstrating the local organ shortage that certainly contributed to the deaths while waiting for the procedure. The patients who died while on the list had waited an average of 63.5–80 days, showing that they could never withstand a 200-day waiting time.

Starting new LT programs in poor regions increases the organ procurement and the number of surgeries. However, the initial graft and patient survival rates can be low because the learning curve tends to be long in complex surgeries such as LT. Addeo et al. [16] identified three learning-curve phases of surgeons performing LT. The first phase lasts through the first 70 procedures and the second from the 71st to the 101st ones. Only after this point, when the surgical time and the need for red cell transfusion become stable, is the third phase achieved. Similarly, Guerra et al. [7] identified significantly higher mortality among patients undergoing the first 30 LTs performed at their institution.

In addition to the technical skills, LT outcomes also depend on donor and recipient conditions [17]. Appropriate combinations of donor and recipient risk factors promote high survival rates for patients and grafts [18]. Thus, the assistance of an experienced transplant team at the time of donor allocation is important for obtaining good results, especially when starting a new LT program. The present study demonstrated that a new LT center can achieve survival results similar to other experienced transplant centers in the same region. Obviously, patients who suffered fulminant hepatitis or those who were prioritized for transplantation were excluded from this comparison, due to their higher mortality rates and absence of similar cases among our patients.

Since the multidisciplinary approach is the key to achieving good outcomes in LT patients, members of the LT department from HCFMUSP made many visits to the new center before starting the new program. In each visit, they were received by the intensive care, nursing, physical therapy and psychology teams in HCFMB. These teams also visited the experienced team facilities, meeting the same professionals involved in the LT. These two-way visits allowed the teams to foresee some barriers that needed to be worked around, such differences between the equipment available in each facility. A big advantage was that the new center team was already

multidisciplinary, composed of all professionals involved in LT because HCFMB had had an LT program some years ago and had an ongoing kidney transplantation program carried out by these professionals. It was crucial for the good results achieved.

Even after the end of the partnership, postoperative mortality at 30 days was similar to that of transplant centers in the same region. The warm ischemia (defined by the period elapsed to perform the liver implant after the placement of the organ in the abdominal cavity) remained stable before and after the partnership between the institutions. To date, no HCFMB patient was relisted because of primary graft dysfunction or arterial thrombosis, suggesting that the new center maintains high-quality skills.

Several other advantages remained after the partnership between the institutions, such as the exchange of patients from one center to another. It is quite common that patients from the countryside of the state seek medical care in major cities, crowding their transplant waiting lists and delaying the surgeries. Since the partnership was established, the HCFMUSP team has referred patients who live in the HCFMB region to the new transplantation center. This practice shortens the distance between these patients and the hospital when they are summoned to receive the LT. Likewise, complex transplant cases, such as recipients with multiple abdominal surgeries or extensive portomesenteric thrombosis are referred from HCFMB to HCFMUSP, after explaining to the patients and their families that some situations require centers that are more experienced. Referring the complex cases and discussing them with the HCFMUSP team have been advantageous for the new center team, which can increase the LT number with less complicated cases and gain experience in the difficult ones without any negative impact on survival rates.

One of the study limitations is that the patient and graft survival rates were evaluated in a short period, especially in relation to the surgeries performed after the end of the partnership. Even so, the number of patients is sufficient to demonstrate that the results were similar to the ones from other transplant centers in the same region. Of note, the absence of primary non-function or arterial thrombosis in the new center and the maintenance of a similar time of warm ischemia also suggest that the partnership reduced the harmful effects of the initial learning-curve period.

## Conclusion

In conclusion, the present study demonstrates that a partnership between universities is a good strategy to start a new LT program, thus diminishing the negative effects reported while the new center is in the initial phases of the LT learning curve. Moreover, it strengthens the exchange of knowledge and patients between the transplant programs, thus increasing organ procurement and helping patients who live far away from major cities. Therefore, this strategy can serve as a model for starting future transplantation programs.

## Acknowledgments

The authors wish to thank all members of the Botucatu Faculty of Medicine (HCFMB) and the Digestive Organs Transplantation Division of the Clinical Hospital of the São Paulo University (HCFMUSP) who have worked in this partnership between the transplantation teams.

## Author Contributions

**Conceptualization:** Rafael Soares Pinheiro, Wellington Andraus, Rodrigo Bronze de Martino, Liliana Ducatti, Rubens Macedo Arantes, Leonardo Pelafsky, Walmar Kerche de Oliveira, Luiz Augusto Carneiro-D'Albuquerque.

**Data curation:** Rafael Soares Pinheiro, Wellington Andraus, Rodrigo Bronze de Martino, Liliana Ducatti, Rubens Macedo Arantes, Leonardo Pelafsky, Claudia Nishida Hasimoto, Fabio da Silva Yamashiro, Lucas Souto Nacif, Luciana Bertocco de Paiva Haddad, Vinicius Rocha Santos, Daniel Reis Waisberg, Matheus Fachini Vane.

**Formal analysis:** Rafael Soares Pinheiro, Wellington Andraus, Liliana Ducatti, Leonardo Pelafsky, Luciana Bertocco de Paiva Haddad, Luiz Augusto Carneiro-D'Albuquerque.

**Investigation:** Rafael Soares Pinheiro, Rubens Macedo Arantes, Leonardo Pelafsky, Claudia Nishida Hasimoto.

**Methodology:** Rafael Soares Pinheiro, Wellington Andraus, Fabio da Silva Yamashiro, Joel Avancini Rocha-Filho, Walmar Kerche de Oliveira.

**Project administration:** Wellington Andraus, Luiz Augusto Carneiro-D'Albuquerque.

**Supervision:** Wellington Andraus, Luiz Augusto Carneiro-D'Albuquerque.

**Writing – original draft:** Rafael Soares Pinheiro, Fernando Gomes Romeiro, Rodrigo Bronze de Martino, Liliana Ducatti, Rubens Macedo Arantes, Leonardo Pelafsky, Claudia Nishida Hasimoto, Fabio da Silva Yamashiro, Walmar Kerche de Oliveira.

**Writing – review & editing:** Wellington Andraus, Fernando Gomes Romeiro, Vinicius Rocha Santos, Joel Avancini Rocha-Filho, Walmar Kerche de Oliveira, Luiz Augusto Carneiro-D'Albuquerque.

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
