## [Decision Letter · Decision Letter 0]

26 Jan 2022

PONE-D-21-39998Model for establishing a new liver transplantation center through mentorship from a University with transplantation expertisePLOS ONE

Dear Dr. Romeiro,

Thank you for submitting your manuscript to PLOS ONE. After careful consideration, we feel that it has merit but does not fully meet PLOS ONE’s publication criteria as it currently stands. Therefore, we invite you to submit a revised version of the manuscript that addresses the points raised during the review process.

This is an interesting work with clinical value and practical importance. We recommend the authors to address the issues such as the concern of survival of patients operated on and mentorship program at the new center. Please refer the followings comments for detail.   

We look forward to receiving your revised manuscript.

Kind regards,

Yun-Wen Zheng

Academic Editor

PLOS ONE

Journal Requirements:

"This study was funded by São Paulo Research Foundation (FAPESP; grant no. 2017/25592-9). FGR received funding from Conselho Nacional de Desenvolvimento Científico e Tecnológico (CNPq)."

Reviewers' comments:

Reviewer's Responses to Questions

**Comments to the Author**

1. Is the manuscript technically sound, and do the data support the conclusions?

Reviewer #1: Partly

Reviewer #2: Yes

2. Has the statistical analysis been performed appropriately and rigorously? 

Reviewer #1: Yes

Reviewer #2: Yes

3. Have the authors made all data underlying the findings in their manuscript fully available?

Reviewer #1: Yes

Reviewer #2: Yes

4. Is the manuscript presented in an intelligible fashion and written in standard English?

Reviewer #1: Yes

Reviewer #2: Yes

5. Review Comments to the Author

Reviewer #1: The authors insist that a partnership between universities is a good strategy to start a new LT program, thus diminishing the negative effects reported while the new center is in the initial phases of the LT learning curve. Moreover, it strengthens the exchange of knowledge and patients between the transplant programs, thus increasing organ procurement and helping patients who live far away from major cities. Therefore, this strategy can serve as a model for starting future transplantation programs.

I have some comments

1. Survival of patients operated on at the new center during the partnership, compared to the survival of patients operated on at other transplant centers outside of the São Paulo State capital in the same time span is different. Liver transplants performed at facilities other than São Paulo tend to have more perioperative deaths. Can you explain the difference from the new facility?

2.Survival of patients operated on at the new center during the first year of the partnership, compared to that of patients operated on at other transplant centers outside of the São Paulo state capital also makes different. The difference in the first month is also noticeable in Figure 2. What are the possible differences between the two?

3. I understand that the comparison between tables 1 and 2 is that there is no big difference between the partnership period and afterwards. I think it is easier to understand if there is comparison data with the high volume center near São Paulo.

4.The description of surgery is ambiguous, such as how surgeons collaborate with each other, how much the doctors at the collaborative facility were actually involved in the surgery, and whether they simply shared the protocol.

5.Once it becomes clear why surgery-related mortality rates are lower at facilities near São Paulo, I feel that improving the protocol of existing facilities is more beneficial than building new ones. In that sense, it would be nice to clarify how much the protocol differs between the existing facility and the new facility. Is it just a difference in surgeon training?

Reviewer #2: The present study demonstrates that a partnership between universities is a good strategy to start a new LT program, thus diminishing the negative effects reported while the new center is in the initial phases of the LT learning curve. Furthermore, it strengthens the exchange of knowledge and patients between the transplant programs, thus increasing organ procurement and helping patients who live far away from major cities. Therefore, this strategy can serve as a model for starting future transplantation programs.

In total, this work is full of clinical value and practice importance, it deserves the potential publication.

Herein, I have a suggestion for the mentorship program. LT is a multidisciplinary project referring to surgery, anesthesia, intensive care medicine, nursing and phychology, and a well organized team in these fields is a make or break condition for LT. Therefore, before LT is performed in the new center, such a team should go to the experienced center for visiting study during mentorship program. That is, the visiting study should be a necessary part of the mentorship program. I strongly suggested the authors add this issue into their study as a drawback.

6. PLOS authors have the option to publish the peer review history of their article (what does this mean?). If published, this will include your full peer review and any attached files.

Reviewer #1: **Yes: **Soichiro Murata

Reviewer #2: No

---

## [Author Response · Author response to Decision Letter 0]

2 Mar 2022

Reviewer #1: The authors insist that a partnership between universities is a good strategy to start a new LT program, thus diminishing the negative effects reported while the new center is in the initial phases of the LT learning curve. Moreover, it strengthens the exchange of knowledge and patients between the transplant programs, thus increasing organ procurement and helping patients who live far away from major cities. Therefore, this strategy can serve as a model for starting future transplantation programs.

I have some comments

1. Survival of patients operated on at the new center during the partnership, compared to the survival of patients operated on at other transplant centers outside of the São Paulo State capital in the same time span is different. Liver transplants performed at facilities other than São Paulo tend to have more perioperative deaths. Can you explain the difference from the new facility?

We thank Reviewer #1 for his/her comments. All of them brought important contributions to our manuscript. Indeed, the survival rates seem dissimilar, but the comparison showed that the difference is not statistically significant and could have occurred by chance. Survival depends on many factors such as the patient’s characteristics, technical skills, equipment available, as well as pre- and postsurgical medical therapy. Any of these factors could have contributed to a trend for better results in the new center, but we cannot conclude it according to the methods adopted. Therefore, we have to consider the results similar in both populations assessed, with no differences or trends between them. We also cannot explain the perioperative deaths from the other centers, because we can only access the survival rates.

2. Survival of patients operated on at the new center during the first year of the partnership, compared to that of patients operated on at other transplant centers outside of the São Paulo state capital also makes different. The difference in the first month is also noticeable in Figure 2. What are the possible differences between the two?

We thank the reviewer for his/her precise observations. Again, the survival rates seem different, but the statistical analysis showed that this difference is not significant. We agree that the data presented could contain a trend, which probably occurred due to the numerical difference between the populations evaluated. Maybe a trend could be observed if we had performed more surgeries in the new center within this time span. However, the only conclusion we can present after a strict analysis of the current data is that the results are similar, showing that the new center achieved survival rates that were not inferior to the ones in the same region.

3. I understand that the comparison between tables 1 and 2 is that there is no big difference between the partnership period and afterwards. I think it is easier to understand if there is comparison data with the high volume center near São Paulo.

This is an excellent idea to be developed in a new study. However, we have no access to data from another center to compare them with the ones obtained in the new center. Only data for the whole country or its regions are available.

4.The description of surgery is ambiguous, such as how surgeons collaborate with each other, how much the doctors at the collaborative facility were actually involved in the surgery, and whether they simply shared the protocol.

We agree with the reviewer and changed this part of the manuscript, giving more details about this point. The changes are highlighted in yellow to be easily found in the text, as shown below:

“An HCFMUSP team composed of a scrub nurse, surgeons and anesthesiologists participated in the recipient's surgery as tutors, always accompanied by an HCFMB surgeon. In the first months, the HCFMB surgeon performed parts of the transplantation, such as the hepatic vein anastomosis, the hepathectomy and other important tasks, which were frequently alternated between HCFMB and HCFMUSP surgeons, encouraging them to share the main tips and concerns. During this period, the experienced surgeons guided the new center team to do the procedures in the best way they knew. As soon as the mentors observed that HCFMB surgeons had done all of these parts safely and rapidly, new ones were shared, until the entire surgery could be carried out by the new team, always under supervision by an HCFMUSP surgeon“.

5.Once it becomes clear why surgery-related mortality rates are lower at facilities near São Paulo, I feel that improving the protocol of existing facilities is more beneficial than building new ones. In that sense, it would be nice to clarify how much the protocol differs between the existing facility and the new facility. Is it just a difference in surgeon training?

This is a remarkable comment and we thank the reviewer for pointing it out. Unfortunately, we have no access to protocols adopted in different liver transplantation centers, so we could not make comparisons between them. We are sure that the effort to make our results available will be followed by doctors from other liver transplantation centers, providing details about strategies they utilize to increase survival rates in their centers and in the new ones that they helped to set up.

Reviewer #2: The present study demonstrates that a partnership between universities is a good strategy to start a new LT program, thus diminishing the negative effects reported while the new center is in the initial phases of the LT learning curve. Furthermore, it strengthens the exchange of knowledge and patients between the transplant programs, thus increasing organ procurement and helping patients who live far away from major cities. Therefore, this strategy can serve as a model for starting future transplantation programs.

In total, this work is full of clinical value and practice importance, it deserves the potential publication.

Herein, I have a suggestion for the mentorship program. LT is a multidisciplinary project referring to surgery, anesthesia, intensive care medicine, nursing and psychology, and a well organized team in these fields is a make or break condition for LT. Therefore, before LT is performed in the new center, such a team should go to the experienced center for visiting study during mentorship program. That is, the visiting study should be a necessary part of the mentorship program. I strongly suggested the authors add this issue into their study as a drawback.

We thank Reviewer #2 for his/her comments and we totally agree with them. As described in the manuscript, the new center team had already performed LT some years before the mentorship. Thus, when the new center team received the experienced team, it was not necessary to teach each step of the surgery, nor the background for the pre- and postsurgical care. Moreover, the new center team visited the HCFMUSP facilities to see how the multidisciplinary team works before beginning the new program. In the same way, the experienced team had visited the HCFMB facilities, discussing each step of the liver transplantation with all the new team members before the surgeries were performed there. It allowed the two teams to foresee some barriers they would have to work out, such as differences between the equipment available in each facility. As we agree with the reviewer that the multidisciplinary work is the key to success, we included a new paragraph in the manuscript discussion. It is highlighted in yellow to make it easy to find it in the text, as follows:

“Since the multidisciplinary approach is the key to achieving good outcomes in LT patients, members of the LT department from HCFMUSP made many visits to the new center before starting the new program. In each visit, they were received by the intensive care, nursing, physical therapy and psychology teams in HCFMB. These teams also visited the experienced team facilities, meeting the same professionals involved in the LT. These two-way visits allowed the teams to foresee some barriers that needed to be worked around, such differences between the equipment available in each facility. A big advantage was that the new center team was already multidisciplinary, composed of all professionals involved in LT because HCFMB had had an LT program some years ago and had an ongoing kidney transplantation program carried out by these professionals. It was crucial for the good results achieved.“

---

## [Decision Letter · Decision Letter 1]

21 Mar 2022

Model for establishing a new liver transplantation center through mentorship from a University with transplantation expertise

PONE-D-21-39998R1

Dear Dr. Romeiro,

We’re pleased to inform you that your manuscript has been judged scientifically suitable for publication and will be formally accepted for publication once it meets all outstanding technical requirements.

Kind regards,

Yun-Wen Zheng

Academic Editor

PLOS ONE

Additional Editor Comments (optional):

Reviewers' comments:

Reviewer's Responses to Questions

**Comments to the Author**

1. If the authors have adequately addressed your comments raised in a previous round of review and you feel that this manuscript is now acceptable for publication, you may indicate that here to bypass the “Comments to the Author” section, enter your conflict of interest statement in the “Confidential to Editor” section, and submit your "Accept" recommendation.

Reviewer #1: All comments have been addressed

Reviewer #2: All comments have been addressed

2. Is the manuscript technically sound, and do the data support the conclusions?

Reviewer #1: Yes

Reviewer #2: Yes

3. Has the statistical analysis been performed appropriately and rigorously? 

Reviewer #1: Yes

Reviewer #2: Yes

4. Have the authors made all data underlying the findings in their manuscript fully available?

Reviewer #1: Yes

Reviewer #2: Yes

5. Is the manuscript presented in an intelligible fashion and written in standard English?

Reviewer #1: Yes

Reviewer #2: Yes

6. Review Comments to the Author

Reviewer #1: The revised manuscript and letter commented on what I pointed out and I'm happy with this edition.I am acceptable in this version.

Reviewer #2: The authors have adequately addressed your comments raised, and I feel that this manuscript is now acceptable for publication.

7. PLOS authors have the option to publish the peer review history of their article (what does this mean?). If published, this will include your full peer review and any attached files.

Reviewer #1: **Yes: **Soichiro Murata

Reviewer #2: No

---

## [Editor Report · Acceptance letter]

22 Mar 2022

PONE-D-21-39998R1 

Model for establishing a new liver transplantation center through mentorship from a University with transplantation expertise 

Dear Dr. Romeiro:

I'm pleased to inform you that your manuscript has been deemed suitable for publication in PLOS ONE. Congratulations! Your manuscript is now with our production department. 

Kind regards, 

on behalf of

Dr. Yun-Wen Zheng 

Academic Editor

PLOS ONE